# Strategies for Pain Management in Hepatocellular Carcinoma Patients Undergoing Transarterial Chemoembolisation: A Scoping Review of Current Evidence

**DOI:** 10.3390/healthcare13090994

**Published:** 2025-04-25

**Authors:** Wei-Zheng Zhang, Kok-Yong Chin, Roshaya Zakaria, Nor Haty Hassan

**Affiliations:** 1Department of Nursing, Faculty of Medicine, Universiti Kebangsaan Malaysia, Kuala Lumpur 56000, Malaysia; p119154@siswa.ukm.edu.my (W.-Z.Z.); roshaya@hctm.ukm.edu.my (R.Z.); 2Department of Pharmacology, Faculty of Medicine, Universiti Kebangsaan Malaysia, Kuala Lumpur 56000, Malaysia; chinky@ukm.edu.my

**Keywords:** hepatocellular carcinoma, transarterial chemoembolisation (TACE), pain management, scoping review

## Abstract

**Background:** Hepatocellular carcinoma (HCC) is a leading cause of cancer-related mortality, with transarterial chemoembolisation (TACE) being a primary treatment for intermediate-stage disease. However, post-procedural pain remains a significant challenge due to inconsistent management practices and a lack of standardised protocols. This scoping review synthesises current evidence on pain management strategies in HCC patients undergoing TACE, evaluates their effectiveness, identifies practice gaps, and proposes optimisation strategies. **Methods**: A comprehensive database search according to the methodological approach given by Arksey and O’Malley with the aid of the PRISMA-ScR guidelines across Cochrane Library, Web of Science, CINAHL, PubMed, and Scopus was performed. The terms associated with pain, TACE, and liver cancer were included in the search strategy. Two independent researchers systematically screened study titles, abstracts, and full texts and extracted key study characteristics and approaches to pain management. **Results**: Of 1515 identified studies, 29 met the inclusion criteria. Most (72.7%) focused on pharmacological interventions, with dexamethasone and lidocaine being the most frequently investigated agents. Non-pharmacological approaches, including psychological interventions, physical therapies, music therapy, health education, and comprehensive nursing, were also reported. Pain was primarily assessed using the visual analogue scale (VAS) and numeric rating scale (NRS). **Conclusions**: Pharmacological interventions, particularly dexamethasone and lidocaine, remain the cornerstone of pain management in TACE, yet consensus on their optimal use is lacking. Non-pharmacological strategies provide complementary benefits. standardised, evidence-based pain management protocols integrating both approaches are needed. Future large-scale, multicentre trials are essential to establish the most effective strategies for optimising patient outcomes.

## 1. Introduction

Hepatocellular carcinoma (HCC) is the most frequent form of cancer arising in the liver and one of the leading causes of cancer death worldwide [1]. It is a malignancy, recurrent and with an increasing pace, and is the third leading cause of death due to cancer in the world [2]. There are many HCC management options, and they depend on several factors, including disease progression, functional capacity, viability of the liver, and the existence of other important medical conditions of the patient. Only if liver cancer is found in its earliest stage are effective curative treatments like liver transplantation or removing the cancer using surgery employed [3]. However, curative treatments are not practical for a large fraction of individuals with intermediate or later-stage diagnoses. Transarterial chemoembolisation (TACE) has been widely accepted as the appropriate therapy for intermediate-stage HCC [4]. TACE is a tumoricidal procedure consisting of the embolisation of the artery followed by the infusion of chemotherapeutic agents directly into the hepatic artery to deliver the cytotoxic to the target tumour and then embolisation to block the artery. This dual action seeks to maximise the local cytotoxic effect on the tumour and minimise systemic exposure and healthy liver tissue [5].

The survival outcomes have been shown to be improved by TACE, and it is known as the treatment approach for individuals not eligible for liver transplantation or surgical tumour removal [6]. Though TACE is effective, it is accompanied by numerous complications, especially severe and impairing adverse effects, of which pain is the most common [7]. After TACE, patients may experience from slight discomfort to severe pain, which may significantly impact their quality of life and treatment experience as a whole [8]. Pain is caused by factors such as anxiety, ischemia damage, and the cytotoxic effects of chemotherapy drugs [9]. Hence, good pain management strategies need to be implemented to improve patient outcomes and increase treatment completion. Current methods for pain relief associated with TACE include the use of medications, including opioids and non-steroidal anti-inflammatory drugs (NSAIDs), as well as non-medical approaches, such as regional anaesthesia and multimodal pain management [10].

Nevertheless, pain management practices are highly variable, and an agreement on optimal protocols does not exist. Due to the important and variable influence of pain on outcomes for HCC patients undergoing TACE, it is crucial to understand the strategies for pain management in HCC patients undergoing TACE. However, previous research on pain management has revealed significant heterogeneity. This gap highlights the necessity for a comprehensive review to synthesise existing pain management strategies. Given the heterogeneity of studies in the literature and in accordance with guidelines [11], a scoping review was undertaken to systematically map existing research, identifying the types of evidence available and research gaps carefully. The scoping review aimed to achieve the following objectives:Systematically map the range of pain management strategies reported in the literature for HCC patients undergoing TACE;Evaluate the reported effectiveness of these strategies in pain management;Identify the commonly used questionnaires or tools in evaluating pain for HCC patients undergoing TACE.

## 2. Materials and Methods

In accordance with Arksey and O’Malley’s methodological framework, this scoping review was conducted. A review protocol was entered into the Open Science Framework (Registered DOI: https://doi.org/10.17605/OSF.IO/9QZ6S, Accessed on 13 March 2025). The studies were undertaken in 5 separate stages: (1) identifying the research question, (2) identifying the relevant studies, (3) study selection, (4) charting data, and (5) collating, summarising and reporting results [12]. This review was reported in accordance with the Preferred Reporting Items for Systematic Reviews and Meta-Analyses (PRISMA) extension for Scopus Reviews (PRISMA-ScR) [13]. The PRISMA-ScR checklist is available in Appendix A.

### 2.1. Identifying the Research Question

Previously published study data have been reviewed to determine if there is anything that can be used in patients with HCC who are receiving TACE to reduce pain. Based on the parameters set for this research, the research question for this study was “What types of pain management are available to control pain for HCC patients undergoing TACE?”.

### 2.2. Identifying the Relevant Studies

A wide variety of sources were identified for this comprehensive scoping review through several research databases to generate pain management strategies for HCC patients who undergo TACE. PubMed, Web of Science, Cochrane Library, Scopus, and CINAHL were thoroughly searched to identify all relevant studies. No restrictions on publication dates were used as no previous review on this specific topic had been performed. The search was last conducted in February 2025. The terms and formulas used for conducting the search were as follows: (“pain” OR “pain management” OR “pain control” OR “analgesia”) AND (“TACE” OR “transarterial chemoembolisation”) AND (“HCC” OR “hepatocellular carcinoma” OR “liver cancer”). The search strategies for each database are available in Appendix A. Studies included in the review must meet the criteria for research purposes. The inclusion criteria were (1) studies with a focus on pain management interventions studied or used in the HCC patients undergoing TACE; (2) studies published in English. The criteria used to exclude the studies were as follows: (1) publications such as books, book chapters, protocols, abstracts, commentaries, editorials, and conference papers; (2) studies where the full text was unavailable; (3) duplicate or closely related articles; (4) case reports; (5) trial registry records; and (6) review articles, including scoping reviews, systematic reviews, and meta-analyses.

### 2.3. Study Selection

Two researchers (WZZ and KYC) conducted an independent review of the titles and abstracts for all identified studies. After initial screening, full-text articles were assessed in terms of their eligibility as per pre-defined criteria. Any disagreement on study selection was discussed and settled. An overview of the screening and article selection process is shown in Figure 1.

### 2.4. Charting Data

A structured form was created to gather comprehensive information regarding a specific set of study characteristics. The extracted data comprised the first author, publication year, country of study, study objectives, design, sample size, type of pain management intervention, outcome measures employed, and key findings related to intervention effectiveness.

### 2.5. Collating, Summarising, and Reporting Results

The data extracted from both researchers were compared and summarised into a single table for analysis. The findings were reported in tables with detailed descriptions of study characteristics and pain management interventions to alleviate the pain of HCC patients undergoing TACE.

## 3. Results

### 3.1. Characteristics of Studies

The most up-to-date pain management strategies for HCC patients undergoing TACE were reviewed. As outlined in Figure 1, 1515 articles were initially found from a database search. Duplicate entries were then removed, and the process was finished with 946 unique articles. Then, of these, the titles and abstracts were screened, and 55 full-text articles were evaluated. After a thorough assessment that was based on pre-defined inclusion and exclusion criteria, we found 29 articles relevant and included them in this scoping review.

Table 1 summarises 29 articles that examined pain management interventions for HCC patients undergoing TACE. Twenty-three of these were found from the comprehensive database search, while a further six were extracted from reference lists of previous meta-analyses. Among those, 41% (12 studies) were randomised controlled trials (RCTs), 55% (16 studies) were cohort studies, and 3% (1 study) was a quasi-experimental crossover study.

Pharmacological interventions were most investigated in the studies, comprising 21 out of 29 studies (72.7%). Three studies (10.3%) examined physical interventions to manage pain, and two studies (6.8%) examined psychological ones. Moreover, some alternative methods of pain management were explored in six studies (10.2%). Dexamethasone was the most often adopted pharmacological agent among the pharmacological treatments used to manage pain in HCC patients undergoing TACE. Table 2 shows that eight studies focused on non-pharmacological methods.

The 29 included studies varied widely in terms of study design, population characteristics, and pain management strategies. These studies were published between 1999 and 2023, with most originating from China. Table 3 provides a detailed summary of the 29 studies that were included in this review.

### 3.2. Pain Assessment Tools

Table 3 presents the assessment tools used in most studies that were included in this scoping review, which were the VAS and NRS. Approximately 10 studies reported using the VAS as a pain level evaluation method in HCC patients treated with TACE [14,15,16,17,18,19,20,21,22,23]. Meanwhile, the NRS was documented in nine studies as a tool for monitoring the pain of patients with HCC who received TACE [10,24,25,26,27,28,29,30,31]. Three studies employed the CTCAE v5.0 system to evaluate pain within the context of treatment-related adverse events [32,33,34]. In addition, two studies used SWOG toxicity coding, which covers abdominal pain in post-TACE adverse event assessment [32,35].
healthcare-13-00994-t003_Table 3Table 3Pain assessment tools utilised in the included studies.ScaleScore RangeAuthorsCTCAEV5.0 (n = 3 studies)Grade 1–5Nitipon et al., 2023 [32]; Lu et al., 2021 [33]; Xu et al., 2019 [34]SWOG toxicity coding (n = 2 studies)0–4 per symptomNitipon et al., 2023 [32]; Panot et al., 2021 [35]Self-developed scale (n = 1 study)
Wassana et al., 2022 [36]VAS (n = 10 studies)0–10Lu et al., 2023 [14]; Sang et al., 2001 [35]; Zeng et al., 2014 [16];Guo et al., 2018 [17];Wang et al., 2021 [18]; Yang et al., 2017 [19]; Mohammad et al., 2014 [20];Zhou et al., 2016 [21]; Quentin et al., 2023 [22]; Yu et al., 2020 [23]NRS (n = 9 studies)0–10Kenji et al., 2019 [24]; Zhou et al., 2012 [25]; Wang et al., 2018 [26];Ning et al., 2016 [27]; Ning et al., 2022 [10]; Gao et al., 2024 [28];Wang et al., 2008 [29]; Lin et al., 2022 [30]; Chang et al., 2020 [31]MDASI-GI (n = 1 study)0–10Xu et al., 2016 [37]CTCAEV5.0 = common terminology criteria for adverse events. SWOG toxicity coding = Southwest Oncology Group toxicity coding score. VAS = visual analogue scale. NRS = numeric rating scale. MDASI-GI = MD Anderson Symptom Inventory–Gastrointestinal module.


### 3.3. Pharmacological Interventions

Dexamethasone was the most studied medication for pain relief, as shown in Table 2. This drug was used in disparate types of patients in disparate clinical situations, including pain management after surgical procedures and pain from invasive procedures. In contrast, other pain-relieving medications such as lidocaine, parecoxib, traditional Chinese medicine, patient-controlled analgesia, and oxycodone were investigated less than others.

Table 4 demonstrates that eight studies from three different countries examined the use of dexamethasone in the pain management of HCC patients undergoing TACE [14,19,24,32,33,35,38,39]. Of these studies (50%), more than half (n = 54, 53%) were conducted in China, with a date of publication reported from 2009 to 2023. All the studies included found that dexamethasone may be a means of dealing with pain for this patient population. Of the six, four were RCTs, and the other two were cohort studies. In one study, dexamethasone was administered intravenously; in five, an intravenous formulation was used. A cohort study [33] in China found that patients treated with dexamethasone had a significantly lower incidence of abdominal pain (36.1%) as compared to those in the TACE-only group (55.6%) (*p* = 0.002). Similarly, another RCT study conducted in South Korea demonstrated that the prophylactic administration of intravenous dexamethasone before chemoembolisation significantly reduced pain [19]. This study supports the pre-emptive use of dexamethasone to mitigate inflammatory responses triggered by TACE, leading to improved patient comfort and reduced analgesic requirements.

Five studies from four different countries documented the use of lidocaine for pain management in HCC patients who underwent TACE (Table 5) [15,18,20,40,41]. Of these studies, the highest proportion (40%) were carried out in the USA, with publication years varying from 1990 to 2021. Within all the included studies, lidocaine was seen to be a useful pain relief option in this patient group. Of them, two were cohort studies, and three were RCTs. Sang [15] performed an RCT in South Korea for an intra-arterial lidocaine administration before and after TACE. The presence of lidocaine administered prior to TACE was observed to be highly effective in the reduction in the presence and the intensity of post-TACE pain. Similarly, the safety and efficacy of intra-arterial buffered lidocaine application before embolisation particle infusion were also assessed during an RCT in Egypt by Mohammad [20]. This study demonstrated that a low dose of 50 mg was sufficient to reduce both peri- and post-procedural pain, as well as decrease the requirement for narcotic analgesics in HCC patients undergoing TACE.

As summarised in Table 6, the investigation of other analgesics was less extensive compared to dexamethasone and lidocaine. Among the less-studied options were parecoxib, traditional Chinese medicine, and oxycodone.

Two studies have been conducted in China regarding the use of traditional Chinese medicine in pain control with TACE in HCC patients. The Jian Pi Li Qi (JPLQ) decoction was proven to afford substantial pain relief in a randomised, double-blind, placebo-controlled study in 140 persons. This study suggests that JPLQ may be a valid alternative to relieve TACE-induced pain in patients with HCC [37]. Similarly, another article, written by Xu in 2019 [34], also claimed that Chaihu-huaji decoction with TACE was also associated with low degrees of post-procedural pain, showing that Chaihu-huaji decoction may be a safe and effective adjunct therapy for pain control for HCC patients who underwent TACE.

Three articles reported the use of parecoxib. An RCT study performed in China [27] confirmed that parecoxib had given perioperatively improved postoperative pain control after TACE. Parecoxib was better than oxycodone and celecoxib for managing pain with a lesser number of adverse effects, but further RCTs [10] showed that parecoxib was the preferred analgesic. Nevertheless, a cohort study [21] found no significant differences in patients’ pain scores, indicating that the pain management results may differ and that there should be customised therapy for pain in TACE patients.

Only one solitary investigation appeared to review the use of oxycodone for pain administration. Zhou et al. [25] evaluated the effectiveness of controlled-release oxycodone in managing the postoperative pain of HCC patients undergoing TACE in an RCT performed in China. The results showed that controlled-release oxycodone provided good pain reduction with an acceptable safety profile. Moreover, it was also found to be cost-effective and a potential pain control alternative for this patient group. Because of these findings, controlled-release oxycodone can be chosen as a reliable analgesic for dealing with TACE-associated pain in HCC patients.

Two studies use the technique of patient-controlled analgesia (PCA), and they are presented in Table 7. A study performed in China [17] reported that the combination of pre-emptive parecoxib and sufentanil-based PCA significantly decreased the pain intensity (all *p* < 0.05) at 2, 4, 6, and 12 h compared to the oral drug combination of tramadol, ketorolac, and acetaminophen. The multimodal group also had fewer cases of postoperative nausea and vomiting and higher patient satisfaction regarding their pain control. Another study performed in France [22] examined a reinforced analgesic protocol (RAP) that incorporated non-opioid and opioid analgesics. Evidence was obtained that RAP reduces (12% vs. 43%; *p* = 0.03) the rate of severe periprocedural abdominal pain (SAP) significantly in patients at high risk of SAP and remarkably decreases the overall use of analgesics. Both studies highlighted the benefits of enhanced PCA strategies for improved pain control in this patient population.

### 3.4. Psychological Interventions

Only two studies examined psychological interventions for managing pain in HCC patients undergoing TACE, and both were completed in China [28,29] (refer to Table 8). In the study of Gao et al. [28], preoperative interviews combined with prospective nursing significantly alleviated patients’ psychological stress reactions. Patients who received psychological treatments exhibited more consistent postoperative vitals and blood pressure patterns, which varied less compared to control patients. Furthermore, they had significantly lower (*p* < 0.05) NRS pain scores at 12, 24, and 48 h postoperatively. Standard medical care combined with psychological interventions led to decreased pain amounts for hepatic arterial chemoembolisation therapy patients, according to Wang et al. [29] (*p* < 0.01). Some psychological factors, including somatisation combined with phobia and anxiety, showed a relationship with pain scores. The results indicated that psychological interventions provided an effective mechanism for reducing pain and enhanced their value as a complementary approach in pharmacological pain management for this patient population.

### 3.5. Physical Interventions

Three studies reported on physical interventions aimed at reducing pain in HCC patients receiving TACE [16,30,31], two of which were RCTs and one of which was a cohort study (Table 9). The intervention of two studies was acupuncture [16,30], and one study was body positing [31]. The first study [16] assessed wrist–ankle acupuncture (WAA) and found that WAA provided comparable pain relief to oral morphine sulphate (MOR) at 1, 2, and 4 h post-intervention (*p* > 0.05) but significantly greater pain relief at 6 h (*p* < 0.05). Additionally, the occurrence of abdominal distension was significantly lower in patients of the WAA group than in the control group (*p* < 0.05). The second study [30] on buccal acupuncture demonstrated a faster onset and longer duration of analgesia compared to TACE alone (*p* < 0.05). It was determined that post-treatment pain scores were significantly lower in the buccal acupuncture group. On top of that, liver function improved, levels of serum tumour markers were reduced, and the patient’s immune function improved with this approach (*p* < 0.05). The third study [31] evaluated body positioning as a pain management intervention for post-TACE back pain. This single-blind RCT revealed that patients who had their positions changed 2 and 4 h after the procedure experienced significantly lower pain scores compared to those who remained in the supine position (*p* < 0.01).

### 3.6. Others (Music Therapy, Health Education, and Comprehensive Nursing)

Music therapy, health education, and comprehensive nursing interventions were reported in three studies [23,26,36] (refer to Table 10). In Thailand, Wassana et al. [36] used a quasi-experimental crossover design to study the effects of music therapy on 30 people who had the intervention of TACE for liver cancer. The results showed a significant decrease in pain levels when music therapy was performed compared to the control period for the first three days (*p* < 0.001, *p* < 0.01). With respect to mild pain (*p* < 0.001), these results confirm the worth of a music therapy intervention. Also, a different study pursued by Wang et al. [26] in 2018 investigated how health education (HE) was associated with pain management among 115 HCC patients. The analysis showed that the intervention group participants had a significantly greater drop in pain levels after HE (*p* < 0.05). In addition, there was an inverse relationship between health-related knowledge and pain-reported intensity (*p* < 0.05). Additionally, a study [23] involving 160 advanced liver cancer patients undergoing TACE found that comprehensive nursing interventions significantly reduced postoperative pain scores compared to routine nursing (*p* < 0.05). These studies demonstrated the effectiveness of non-pharmacological interventions such as music therapy, health education, and comprehensive nursing in pain management for HCC patients undergoing TACE.

For gaining a better understanding of the findings and comparisons, Table 11 combines all the tables with details of the type of scale used, the number of patients included, and the corresponding score of each scale.

## 4. Discussion

This scoping review, which included 29 studies published before February 2025, provides a comprehensive overview of the current pain management strategies for HCC patients undergoing TACE, most of which focus on pharmacological intervention. This review found that there was considerable variability in pain management approaches, highlighting the need for standardised protocols to improve patient outcomes.

### 4.1. Pain Assessment Tools

The frequent use of the VAS and NRS in the included studies reflects their reliability and ease of use in clinical settings. However, the diversity of pain assessment tools used across studies indicates a need for standardised pain assessment methods to ensure consistency in measuring outcomes and comparing the effectiveness of different interventions. While the VAS and NRS are widely accepted, their subjective nature may limit their ability to capture the full spectrum of pain experiences, particularly in diverse cultural contexts or among patients with cognitive impairments. Thus, developing more comprehensive and culturally sensitive assessment tools could enhance the accuracy of pain evaluation and lead to more tailored treatment approaches. However, unlike the VAS and NRS, which directly measure pain intensity, the SWOG system primarily evaluates overall treatment-related toxicity, incorporating symptoms such as fever, nausea, vomiting, and abdominal pain [42]. While it provides some insight into post-procedural discomfort, it is not a dedicated pain assessment tool. Similarly, the CTCAE v5.0 system classifies pain severity into specific grades for the purpose of adverse event reporting rather than assessing pain intensity on a continuous scale. This grading system standardises pain assessment across studies [43]; however, it may not effectively capture subtle changes in pain perception compared to the VAS or NRS.

### 4.2. Pharmacological Interventions

Pharmacological interventions, particularly the use of dexamethasone and lidocaine, dominate the literature, reflecting a reliance on these agents in managing pain associated with TACE. Dexamethasone has consistently shown efficacy in several studies from China [14,19,33,38], suggesting the pivotal role it plays in TACE-related pain management. However, variations in administration routes (intravenous vs. oral) and dosages indicate the necessity for further research to establish the optimal regimen. Although lidocaine was less studied compared to dexamethasone, its potential in multimodal pain management strategies warrants attention. The findings underscore a lack of consensus on the use, dosage, or combination with other analgesics [15,18,20,40,41], indicating a need for more standardised, evidence-based guidelines in the literature. The use of lidocaine may be underutilised, notably in non-invasive form, and thus a window of opportunity to improve patient comfort and pain relief may be missed. Further exploration of the role of lidocaine in future studies may be of worth, possibly in combination with newer analgesic agents or as part of the multimodal pain management protocols.

The insufficient evidence to support the broad use of other analgesics (parecoxib, corticosteroids, oxycodone) may suggest underutilisation in clinical practice. Corticosteroids and parecoxib, in particular, for which inconsistent results have been reported in the literature, need to be rigorously studied in terms of their efficacy in the same patient population [10,17,21,24,27]. Given the growing importance of such individualised (or personalised) medicine, it might be of value to explore the effects of these analgesics in particular patient subpopulations based on their genetic profiles or other co-existing conditions that might modify their way of experiencing pain or how drugs are metabolised.

### 4.3. Non-Pharmacological Interventions

The less commonly reported non-pharmacological interventions provide important adjuncts to conventional pain management. Acupuncture shows promise of pain reduction without the side effects of pharmacological treatments [16,30]. This finding is consistent with a systematic review that indicated that acupuncture may serve as an effective and safe method for pain reduction in the palliative care of cancer patients [44]. Research indicates that acupuncture influences pain through various mechanisms, including the activation of neurotransmitters such as opioids, serotonin, norepinephrine, and endocannabinoids within the central nervous system. Furthermore, acupuncture appears to regulate the hypothalamic–pituitary–adrenal axis, resulting in peripheral opioid release and decreased levels of inflammatory mediators like cyclooxygenase-2 and prostaglandin E2 [45]. These processes may explain the pain relief observed in TACE patients undergoing acupuncture. Nonetheless, additional high-quality RCTs are necessary to establish standardised protocols and assess their long-term efficacy in this particularly patient group.

Similarly, positioning adjustment techniques are also essential in the management of pain for patients undergoing TACE [31]. Since this included study was the first to assess HCC patients’ placement following TACE efficacy and safety, there is inadequate evidence to explain the precise mechanisms through which positioning alleviates pain. However, previous systematic research in other populations, such as newborns [46], indicates that positioning may alleviate pain by pressure redistribution, modulation of the autonomic nervous system, and enhancement of musculoskeletal alignment. Additional research is required to verify the applicability of these mechanisms to TACE patients and to develop standardised clinical guidelines for positioning adjustments in this context.

With the global shift in medicine toward integrative medicine, non-pharmacological interventions should be considered not merely adjuncts but as potential front-line treatments, especially for patients who are likely to experience significant side effects from drug-based therapies.

Music therapy demonstrates potential as a non-pharmacological approach for alleviating pain in HCC patients following TACE [36], which aligns with a recent study that determined music therapy to be an effective intervention for alleviating pain and anxiety in cancer patients [47]. Previous research [48] has explored the mechanisms underlying the pain-relieving effects of music therapy: music helps relax and reduce tension by affecting heart rate and breathing, which may relieve pain. Music therapy also stimulates the limbic system, which releases endorphins and increases parasympathetic nervous system activity, reducing pain. Music may also affect pain perception and emotions via regulating central nervous system activity in the brainstem. However, future studies are necessary to validate these mechanisms and assess the long-term efficacy of music therapy in pain management for HCC patients receiving TACE.

Health education is essential for enabling patients to effectively manage their pain [26]. Health education can alleviate anxiety and improve patients’ coping mechanisms by offering information on pain management strategies, the treatment process, and potential side effects [49]. In the context of TACE, educating patients regarding their condition and treatment options may enhance their confidence and improve overall quality of life. Health education effectively reduces pain perception by addressing patients’ fears and misconceptions regarding treatment.

Nursing care is essential in the management of pain, especially for cancer patients receiving TACE [28]. Nurses serve as the initial point of contact for patients, playing a crucial role in evaluating pain levels, implementing pain relief strategies, and offering emotional support [50]. Effective communication and personalised care plans are crucial for optimising pain management and ensuring that patients receive appropriate treatment. The comprehensive approach to pain management implemented by nurses can enhance patient outcomes and overall quality of life.

The ability of psychological interventions, however, further emphasises the significance of the psychological aspect of pain [29]. Reductions in pain scores, along with an improvement in the overall well-being of the patient, suggest that these interventions can complement pharmacological treatments, thus providing more comprehensive pain management. In line with the increasing trend towards the acceptance of the biopsychosocial model of pain [51], psychological factors can be considered to be relevant factors in the perception of and coping with pain. These non-pharmacological strategies can be combined with standard care and may assist in lowering the needed dose of analgesics in general, perhaps decreasing adverse effects and enhancing the patient experience.

### 4.4. Implications for Clinical Practice

This review has several implications for clinical practice. First and foremost, there is a greater need for the development of standardised (pharmacological as well as non-pharmacological) pain management protocols [52]. Considering the variability in current practices, setting evidence-based guidelines will be essential for the optimisation of pain control and better outcomes in patients [53]. In addition, tailoring pain management strategies on the individual patient also needs to be based on the context of the individual patient, i.e., the cultural background of the individual patient, the psychological state of the individual patient, and previous pain experiences of the individual patient [54]. Since patients in the HCC who undergo TACE have different needs, a one-size approach may not adequately address their needs.

Second, other strategies should be regarded as essential parts of pain treatment protocols, particularly so for patients who may be liable to side effects from pharmacological treatment [55]. Together, psychological support and physical interventions may increase the effectiveness of both pain management. Furthermore, it may also be prudent to investigate the impact of patient education on pain management because educated patients will be more interested and involved in their care and more likely to follow treatment plans [56]. Information regarding the advantages and limitations of different pain control approaches could supply patients with the ability to participate proactively in the decision-making process to attain better pain control and satisfaction with care.

Finally, additional research should concentrate on large-scale multicentre trials to discover optimal pain management combinations [57]. Such studies will help bridge the gap in the currently available literature and will guide clinicians in the management of pain in the HCC patient receiving TACE. Patient-reported outcomes should also be prioritised in such trials, beyond just a reduction in pain intensity but also the impacts of the treatment on quality of life, functionality, and psychological well-being [58]. Additionally, patient-centred research approaches can assist in creating pain management protocols that effectively deal with patient needs together with their expectations and preferences.

### 4.5. Limitations

Some limitations exist in this scoping review. First, limited inclusion criteria to English publications may have excluded important studies in other languages. Second, most of the research papers are from China, which may reflect regional clinical practices and limit generalisability to other healthcare settings. The paucity of appropriate studies from other countries that matched our inclusion criteria is the main reason for this. These findings are still useful given China’s high hepatocellular carcinoma rate and significant TACE-related pain management experience. More international studies should be included in future studies to improve generalisability. Third, the studies differ in design, sample size, and interventions, making it hard to determine the best pain management options. Notably, this review includes both RCTs and observational studies like cohort studies, which vary in evidence strength. Scoping reviews rarely assess study quality, but noting these variations helps avoid misinterpretation. Future systematic reviews and meta-analyses with explicit risk of bias evaluations could build on this foundation for more conclusive findings. Standardising study methods, including pain assessment and reporting, allows for relevant comparisons and high-quality evidence synthesis.

## 5. Conclusions

In conclusion, this scoping review summarised current pain management strategies for HCC patients undergoing TACE, most of which focus on pharmacological intervention. Pharmacological and non-pharmacological interventions integrated into a single pain management programme tailored to the needs of each patient provide considerable potential for improving the overall outcomes of the targeted population. Despite the progress in understanding pain management for TACE patients, several gaps remain. Few studies directly compared different pain management strategies, and there is limited high-quality evidence from RCTs. Moreover, the long-term outcomes of pain management interventions and their impact on overall patient prognosis were seldom addressed. Further research is needed to establish standardised protocols for pain management in these patients.

## Figures and Tables

**Figure 1 healthcare-13-00994-f001:**
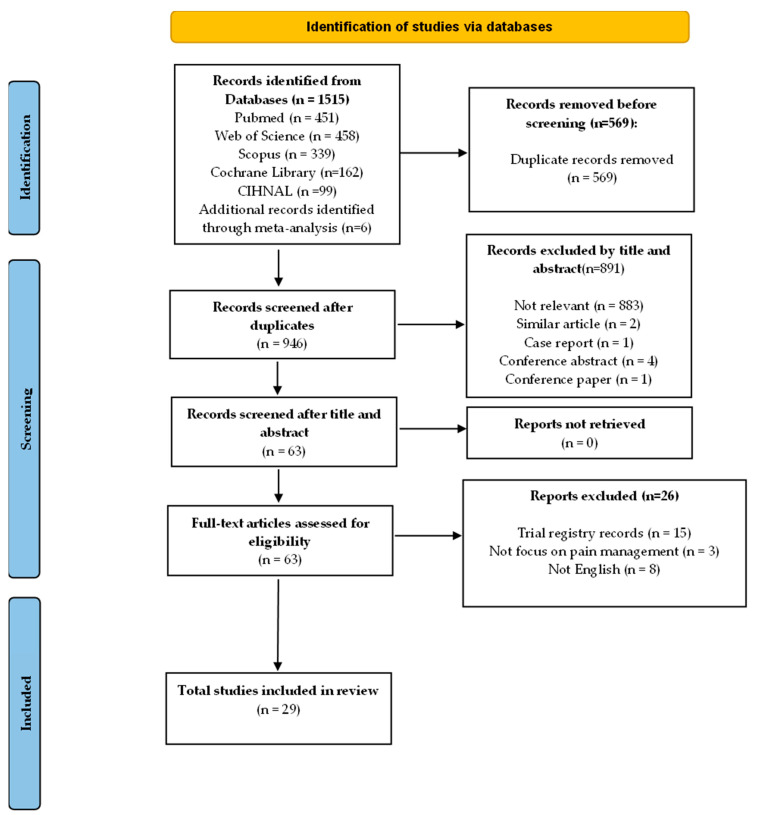
Flowchart of PRISMA applying to the study selection process.

**Table 1 healthcare-13-00994-t001:** Study designs and different categories of intervention for the selected studies.

Design	N (%)
Cohort study	16 (55%)
Quasi-experimental crossover study	1 (3%)
Randomised controlled trial	12 (41%)
Category of intervention	
Pharmacological	21 (72%)
Physical	3 (10%)
Psychological	2 (7%)
Others	3 (10%)

**Table 2 healthcare-13-00994-t002:** Pain management interventions in the 29 included studies (some studies included multiple interventions).

Intervention	N (%)
Medication	
Dexamethasone	8 (28.0%)
Lidocaine	5 (17.2%)
Chinese medicine	2 (6.9%)
Patients control analgesia	2 (6.9%)
Oxycodone	1 (3.4%)
Parecoxib sodium	3 (10.3%)
Psychological	
Preoperative interview and prospective nursing	1 (3.4%)
Psychological intervention	1 (3.4%)
Physical	
Acupuncture	2 (6.9%)
Body positing	1 (3.4%)
Others	
Music therapy	1 (3.4%)
Health education	1 (3.4%)
Comprehensive nursing	1 (3.4%)

**Table 4 healthcare-13-00994-t004:** Research studies investigating the use of dexamethasone for pain management in HCC patients undergoing TACE.

Author(s)	Country	Year	Study Design	Medication	Pain Description (Intervention vs. Control)	Main Findings
Nitipon et al. [32]	Thailand	2023	RCT	Dexamethasone and N-acetylcysteine	I: 1% C: 18%	Intravenous dexamethasone and N-acetylcysteine combination ameliorated the occurrence of pain after cTACE in patients with intermediate-stage HCC.
Lu et al. [33]	China	2021	Cohort Study	Dexamethasone	I: 44 (36.1%) C: 74 (55.6%)	TACE-only group vs. TACE + intravenous dexamethasone group: incidence of abdominal pain—74 patients (55.6%) vs. 44 patients (36.1%), *p* = 0.002.
Panot et al. [35]	Thailand	2021	RCT	Dexamethasone	I: 36% C: 56%	Pain levels were reduced in the intravenous dexamethasone group compared to the placebo group (*p* = 0.16), with an incidence of 56% versus 36%.
Lu et al. [14]	China	2023	Cohort Study	Dexamethasone and palonosetron	I: 40.6% C: 56.6%	The combined use of intravenous dexamethasone and palonosetron before D-TACE can effectively reduce the incidence of pain.
Feng et al. [38]	China	2009	RCT	Dexamethasone and ginsenosides	I: 20% C: 80%	A combination of oral intake dexamethasone and ginsenosides can control pain for patients following TACE.
Yang et al. [19]	South Korea	2017	RCT	Dexamethasone	I: 2.2 ± 2.52 C: 3.5 ± 3.22	The prophylactic administration of intravenous dexamethasone before chemoembolisation is an effective way to reduce patients’ pain.
Kenji et al. [24]	Japan	2019	Cohort study	Steroids	I: 5.8 ± 7.3 C: 5.8 ± 8.7	This study demonstrates that steroids were ineffective at preventing pain following TACE in patients with HCC.
Matthew et al. [39]	USA	2013	Cohort study	Steroids	I: 2.4 ± 2.6 C: 2.4 ± 2.2	The use of steroids did not affect analgesic agent use

**Table 5 healthcare-13-00994-t005:** Research studies investigating the use of lidocaine for pain management in HCC patients undergoing TACE.

Author(s)	Country	Year	Study Design	Main Findings
Sang et al. [15]	South Korea	2001	RCT	Administering lidocaine intra-arterially before TACE is significantly more effective than post-TACE administration in lowering both the occurrence and intensity of post-procedural pain.
Molgaard et al. [41]	USA	1990	Cohort Study	Significant decreases in the use of narcotic analgesics during HCE procedures can be safely accomplished through the administration of intra-arterial lidocaine.
Wang et al. [18]	China	2021	RCT	Intra-arterial lidocaine via the water-in-oil technique is a safe and effective treatment of both intra-procedural and post-TACE pain.
Hartnell et al. [40]	USA	1999	Cohort Study	Lidocaine administered intra-arterially during chemoembolisation decreases the intensity and duration of pain following the procedure.
Mohammad et al. [20]	Egypt	2014	RCT	Intra-arterial administration of buffered lidocaine before infusing the embolisation particle of TACE is safe and effective in doses as low as 50 mg for reducing peri- and post-procedural pain and dosage of narcotic analgesics in patients with HCC.

**Table 6 healthcare-13-00994-t006:** Research studies examining the use of traditional Chinese medicine, parecoxib, and oxycodone for pain management in HCC patients undergoing TACE.

Name	Author(s)	Country	Year	Study Design	Main Findings
Chinese medicine (n = 2)	Xu et al. [37]	China	2016	RCT	JPLQ decoction may be an effective modality to relieve pain after TACE.
	Xu et al. [34]	China	2019	Cohort study	Chaihuhuaji decoction can reduce patients’ pain after TACE.
Parecoxib (n = 3)	Ning et al. [27]	China	2016	RCT	Using parecoxib in the perioperative phase dramatically improved its ability to suppress pain following TACE, making it a better treatment for post-procedural pain.
	Ning et al. [10]	China	2022	RCT	Compared to oxycodone and celecoxib, parecoxib provides better pain management along with a lower adverse effect profile.
	Zhou et al. [21]	China	2016	Cohort study	There were no differences in pain scores.
Oxycodone (n = 1)	Zhou et al. [25]	China	2012	RCT	Controlled-release oxycodone appears to present a secure, efficient, and inexpensive alternative to control postoperative pain following TACE in patients with unresectable liver cancer.

**Table 7 healthcare-13-00994-t007:** Studies reporting the use or evaluation of patient-controlled analgesia.

Author(s)	Country	Year	Study Design	Main Findings
Guo et al. [17]	China	2018	Cohort study	It has been demonstrated that a multimodal analgesia approach involving pre-emptive sufentanil and parecoxib is a safe, efficacious, and cost-effective strategy for the management of postoperative pain in HCC patients undergoing TACE.
Quentin et al. [22]	France	2023	Cohort study	Effective use of both opioid and non-opioid analgesics combined with enhancing existing pain management protocol thoroughly decreases perioperative pain in patients undergoing HCC who underwent conventional TACE.

**Table 8 healthcare-13-00994-t008:** Studies reporting the use or evaluation of psychological interventions.

Author(s)	Country	Year	Combined with Medication	Study Design	Main Findings
Gao et al. [28]	China	2024	No	RCT	Preoperative visits and prospective nursing interventions can effectively relieve patients’ pain.
Wang et al. [29]	China	2008	Yes (specific medication is not mentioned)	RCT	Psychological treatments used in combination with standard medications reduce pain during hepatic arterial chemoembolisation procedures and should be integrated as part of pain care strategies.

**Table 9 healthcare-13-00994-t009:** Studies reporting the use or evaluation of physical intervention.

Author(s)	Country	Year	Study Design	Main Findings
Zeng et al. [16]	China	2014	RCT	Wrist–ankle acupuncture has an analgesic effect equal to or greater than oral morphine sulphate in HCC patients with moderate to severe post-TACE pain.
Lin et al. [30]	China	2022	Cohort study	Buccal acupuncture can reduce the degree of pain and liver function damage in patients with advanced-stage primary liver cancer.
Chang et al. [31]	Taiwan, China	2020	RCT	Changing patients’ body positions in bed after transcatheter arterial chemoembolisation is a safe and effective method of decreasing back pain. Clinicians should change the positions of people with hepatocellular carcinoma 2 h after they receive transcatheter arterial chemoembolisation.

**Table 10 healthcare-13-00994-t010:** Research on implementation or assessment of alternative interventions.

Author(s)	Country	Year	Study Design	Main Findings
Wassana et al. [36]	Thailand	2022	Quasi-experimental crossover study	Music therapy effectively reduces mild pain among patients with primary liver cancer undergoing TACE.
Wang et al. [26]	China	2018	RCT	Health education in HCC patients before TACE improves pain management during the procedure.
Yu et al. [23]	China	2020	Cohort study	Comprehensive nursing can reduce postoperative pain.

**Table 11 healthcare-13-00994-t011:** Summary of pain assessment tools, patient numbers, and pain scores across included studies.

Author and Year (Country)	Study Design	Sample Size	Tool	Pain Results (Mean ± SD) (Control vs. Intervention)
Nitipon et al., 2023 (Thailand) [32]	RCT	100	SWOG	4.04 ± 2.2 vs. 0.38 ± 1.1
Lu et al., 2021 (China) [33]	Cohort study	255	CTCAEV5.0	55.6% vs. 36.1%
Panot et al., 2021 (Thailand) [35]	RCT	100	SWOG	3.71 vs. 2.14
Wassana et al., 2022 (Thailand) [36]	Quasi-experimental crossover study	30	Self-developed the severity of the PES symptom scale	2.93 ± 1.36 vs. 2.00 ± 1.08
Lu et al., 2023 (China) [14]	Cohort study	278	VAS	3.2 ± 1.1 vs. 2.9 ± 0.8
Kenji et al., 2019 (Japan) [24]	Cohort study	144	NRS	5.8 ± 8.7 vs. 5.8 ± 7.3
Sang et al., 2001 (South Korea) [15]	RCT	113	VAS	4.9 ± 2.0 vs. 3.1 ± 2.8
Feng et al., 2009 (China) [38]	RCT	120	Incidence of pain	80% vs. 20%
Xu et al., 2016 (China) [37]	RCT	150	MDASI-GI	37% vs. 7%
Zeng et al., 2014 (China) [16]	RCT	60	VAS	NA
Guo et al., 2018 (China) [17]	Cohort study	84	VAS	NA
Zhou et al., 2012 (China) [25]	RCT	210	NRS	4.8 ± 1.2
Molgaard et al., 1990 (USA) [41]	Cohort study	45	Need for analgesics	11.7 ± 4.4 vs. 0.13 ± 0.89
Wang et al., 2021 (China) [18]	RCT	70	VAS	NA
Hartnell et al., 1999 (USA) [40]	Cohort study	56	Need for analgesics	69% vs. 19%
Wang et al., 2018 (China) [26]	RCT	115	NRS	NA
Ning et al., 2016 (China) [27]	RCT	120	NRS	3.1 ± 1.2 vs. 1.1 ± 0.8
Ning et al., 2022 (China) [10]	RCT	312	VAS	4.40 ± 2.85 vs. 1.26 ± 1.73
Yang et al., 2017 (South Korea) [19]	RCT	88	VAS	3.5 ± 3.22 vs. 2.2 ± 2.52
Matthew et al., 2013 (USA) [39]	Cohort study	125	Need for analgesics	2.4 ± 2.5 vs. 2.0 ± 2.2
Mohammad et al., 2014 (Egypt) [20]	RCT	39	VAS	7.4 ± 1.2 vs. 3.2 ± 1.1
Gao et al., 2024 (China) [28]	RCT	86	NRS	NA
Wang et al., 2008 (China) [29]	RCT	262	NRS	1.64 ± 1.53 vs. 0.29 ± 0.21
Zhou et al., 2016 (China) [21]	Cohort study	242	VAS	0.83 ± 1.58 vs. 0.69 ± 1.66
Lin et al., 2022 (China) [30]	Cohort study	80	NRS	2.52 ± 0.38 vs. 1.68 ± 0.27
Quentin et al., 2023 (France) [22]	Cohort study	83	VAS	7 ± 16 vs. 3 ± 12
Chang et al., 2020 (Taiwan, China) [31]	RCT	78	NRS	2.97 ±2.22 vs. 0.97 ±1.42
Yu et al., 2020 (China) [23]	Cohort study	160	VAS	5.86 ± 0.78 vs. 5.57 ± 0.82
Xu et al., 2019 (China) [34]	Cohort study	125	CTCAEV5.0	34% vs. 29%

## Data Availability

Not applicable.

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
