# Peer review of "Strategies for Pain Management in Hepatocellular Carcinoma Patients Undergoing Transarterial Chemoembolisation: A Scoping Review of Current Evidence"

_healthcare, 2025, doi:10.3390/healthcare13090994_

Round 1

Reviewer 1 Report

Comments and Suggestions for Authors

Lines

Item

Correction

All

The manuscript inconsistently refers to patient groups as "PLC" (Primary Liver Cancer) and "HCC" (Hepatocellular Carcinoma) interchangeably.

To be uniform

Line 75:

 “necessity for a a comprehensive review” → Typographical error: duplicated “a”.

References

In the body text, several references are cited by brackets (e.g., [14], [15]), but the bibliography formatting is inconsistent.

References

The citations are listed at the end without consistent alignment with the in-text citations. Some entries lack full reference metadata (e.g., journal names, volume, page numbers).

References

For instance, [14] and [15] in-text are discussed as pivotal dexamethasone studies, but their entries are embedded in dense blocks without proper formatting for easy verification.

                            References and discussion

the high number of studies from China, which may skew findings toward regional practices not generalizable globally.

Discussion

While the review acknowledges music therapy, acupuncture, psychological interventions, and nursing care, these are under-analyzed.

Only 8 out of 29 studies focused on non-drug strategies, and these sections are notably shorter despite promising findings.

Study design

The authors state that "all the studies in this review were an experimental research design" (line 153), but this is incorrect.

Cohort studies are observational, not experimental.

Only RCTs and the one quasi-experimental study are experimental by design.

The study design

Scoping reviews often don’t assess study quality by design (per Arksey & O’Malley’s framework), but the lack of any comment on risk of bias in cohort vs. RCT studies may mislead readers about evidence strength.

Results

While methods mention extracting sample size, participant demographics, etc. (line 131), these are mostly absent from the results.

We don’t see average ages, sex breakdowns, or comorbidities across the studies.

Comments on the Quality of English Language

Dear authors 

there are repeating specific words and letter. this may need to be reviewed 

yours 

Author Response

Dear Reviewer 1,

Thank you for your comments. Our response is attached. 

--------------------------------------------------------------------------------

Response to Reviewer 1

Title:

Strategies for Pain Management in Hepatocellular Carcinoma Patients Undergoing Transarterial Chemoembolisation: A Scoping Review of Current Evidence

ID: healthcare-3552969

Dear reviewer,

Thank you for reviewing our manuscript. We appreciate the constructive comments given and have responded to each of them as the following:

Comment

Reply

1. The manuscript inconsistently refers to patientgroups as "PLC" (Primary Liver Cancer) and "HCC"(Hepatocellular Carcinoma) interchangeably.

Thank you for the comment. We acknowledge the inconsistency in terminology and have standardized it by using “HCC” (Hepatocellular Carcinoma) throughout the manuscript.

2. Line 75: “necessity for a a comprehensive review” → Typographical error: duplicated “a”.

Thank you for pointing out this typographical error. We have corrected it by removing the extra "a" in the revised manuscript. (Line 76).

3. References: In the body text, several references are cited by brackets (e.g., [14], [15]), but the bibliography formatting is inconsistent.

Thank you for the comment. We have reviewed and revised the reference formatting to ensure consistency between in-text citations and the bibliography.

4. References: The citations are listed at the end without consistent alignment with the in-text citations. Some entries lack full reference metadata (e.g., journal names, volume, page numbers).

Thank you for the reminder. We have carefully revised the reference list to ensure that all entries include complete metadata (e.g., journal names, volume, page numbers) and that the references are aligned consistently with the intext citations.

5. References: For instance, [14] and [15] in-text are discussed as pivotal dexamethasone studies, but their entries are embedded in dense blocks without proper formatting for easy verification.

Thank you for your comment. We have revised the formatting of the reference list to improve clarity and ensure that pivotal studies such as [14] and [15] are easy to identify and verify.

6. References and discussion: the high number of studies from China, which may skew findings toward regional practices not generalizable globally.

TACE-related pain management experience. More international studies should be included in future study to improve generalisability”

Thank you for the comment. We acknowledge that most of the included studies originate from China, which may influence the generalizability of the findings. However, this is due to the fact that studies from other countries did not meet our inclusion criteria. We have now addressed this as a limitation in the discussion section:Line 466-472: “most of the research are from China, which may reflect regional clinical practices and limit generalisability to other healthcare settings. The paucity of appropriate studies from other countries that matched our inclusion criteria is the main reason. These findings are still useful given China's high hepatocellular carcinoma rate and significant

7. Discussion: While the review acknowledges music therapy, acupuncture, psychological interventions, and nursing care, these are underanalyzed.

Thank you for your valuable comment. We have expanded the discussion on music therapy, acupuncture, psychological interventions, and nursing care.Line 402-435

8. Study design: The authors state that "all the studies in this review were an experimental research design" (line 153), but this is incorrect. Cohort studies are observational, not experimental. Only RCTs and the one quasiexperimental study are experimental by design.

Thank you for pointing out this issue. We have corrected that:Line 149-151: “Among those, 41% (12 studies) were randomised controlled trials (RCTs), 55% (16 studies) were cohort studies, and 3% (1 study) was a quasi-experimental crossover study.”

9. Study design: Scoping reviews often don’t assess study quality by design (per Arksey & O’ Malley’s framework), but the lack of any comment on risk of bias in cohort vs. RCT studies may mislead readers about evidence strength.

Thank you for the comment. We have added that:Line 473-477: “Notably, this review includes both RCTs and observational studies like cohort studies, which vary in evidence strength. Scoping reviews rarely assess study quality, but noting these variations helps avoid misinterpretation.”

10. Results: While methods mention extracting sample size, participant demographics, etc. (line 131), these are mostly absent from the results. We don’t see average ages, sex breakdowns, or comorbidities across the studies.

Thank you for your comment. We have revised the Methods section to clarify that demographic information:Line 127-131: “A structured form was created to gather comprehensive information regarding a specific set of study characteristics. The extracted data comprised the first author, pub-lication year, country of study, study objectives, design, sample size, type of pain management intervention, outcome measures employed, and key findings related to intervention effectiveness.”

Final remarks:

We thank the reviewer for evaluating our manuscript once again. We hope the revised manuscript can meet the standard of the esteemed journal.

Reviewer 2 Report

Comments and Suggestions for Authors
  1. Does the SWOG toxicity coding assess pain, or is it only for toxicity?
  2. Please discuss “CTCAEV5.0” regarding its use in pain assessment
  3. The references in section 3.2 are listed by author name and year, making it difficult to trace these references in the bibliography. Some citations are in superscript, while others are in normal font. Please ensure consistency in citing references throughout the manuscript.
  4. The reference "Nitipon et al., 2023" is not listed in the bibliography.
  5. Table 4: Please include the scores used to assess pain and their corresponding values to better understand the findings.
  6. If medication is used along with psychological and physical interventions, please mention this in the table by adding a separate column. Also, specify if more than one medication is used in the pharmacological intervention.
  7. It would be beneficial to combine all the tables with details of the type of scale used, the number of patients included, and the corresponding score of each scale to gain a better understanding of the findings and comparisons.
  8. Compare all types of pain assessment tools with their score ranges in a table to provide a clearer understanding.
  9. Please clarify PLC and HCC in the manuscript. PLC refers to the broader category of liver carcinomas, whereas HCC is a specific type of PLC.

Author Response

Dear reviewer 2,

Thank you for your comments. Our response is attached. 

----------------------------------------------------------------------------------------------------------

Response to Reviewer 2

Title:

Strategies for Pain Management in Hepatocellular Carcinoma Patients Undergoing Transarterial Chemoembolisation: A Scoping Review of Current Evidence

ID: healthcare-3552969

Dear reviewer,

Thank you for reviewing our manuscript. We appreciate the constructive comments given and have responded to each of them as the following:

Comment

Reply

1. Does the SWOG toxicity coding assess pain, or is it only for toxicity?

Thank you for your commen. we have clarified its role in both the Results and Discussion:Results Line 176-177: “In addition, 2 studies used SWOG toxicity coding, which covers abdominal pain in post-TACE adverse event assessment [33,36].”Discussion Line 343-347: “However, unlike the VAS and NRS scales, which directly measure pain intensity, the SWOG system primarily evaluates overall treatment-related toxicity, incorporating symptoms such as fever, nausea, vomiting, and abdominal pain [43]. While it provides some insight into post-procedural discomfort, it is not a dedicated pain assessment tool.”

2. Please discuss “CTCAEV5.0” regarding its use in pain assessment.

Thank you for the comment. We have added the followin to the discussion:Line 347-351: “Similarly, the CTCAE v5.0 system classifies pain severity into specific grades for the purpose of adverse event reporting, rather than assessing pain intensity on a continu-ous scale. This grading system standardises pain assessment across studies [44]; however, it may not effectively capture subtle changes in pain perception compared to VAS or NRS.”

3. The references in section 3.2 are listed by author name and year, making it difficult to trace these references in the bibliography. Some citations are in superscript, while others are in normal font. Please ensure consistency in citing references throughout the manuscript.

Thank you for the comment. We have standardized the reference format throughout the manuscript, ensuring consistency in citation style and alignment with the reference list.

4. The reference "Nitipon et al., 2023" is not listed in the bibliography.

Thank you for your comment. We have added the reference "Nitipon et al., 2023" to the bibliography:Line 576-577: [33]

5. Table 4: Please include the scores used to assess pain and their corresponding values to better understand the findings.

Thank you for your comment. We have added pain description in Table 4 to provode a clearer understanding of the findings.Line 202 Table 4 pain description

6. If medication is used along with psychological and physical interventions, please mention this in the table by adding a separate column. Also, specify if more than one medication is used in the pharmacological intervention.

Thank you for your comment. We have added a medication column to the table 4 and specified in table 8 whether medication is used along with psychological. Sine no medicaiton is used along with physical interventions, no changes for its part.Line 202 Table 4Line 280 Table 8

7. It would be beneficial to combine all the tables with details of the type of scale used, the number of patients included, and the corresponding score of each scale to gain a better understanding of the findings and comparisons.

Thank you for the suggestion. We have added table 11 to present the type of pain assessment scale used, the number of patients included, and the corresponding scores for each study.Line 325: Table 11 Summary of pain assessment tools, patient numbers, and pain scores across included studies

8. Compare all types of pain assessment toolswith their score ranges in a table to provide aclearer understanding.

Thank you for the comment. We have added the score ranges of all pain assessment tools in Table 3 to provide a clearer understanding, as suggested.Line 178: table 3

9. Please clarify PLC and HCC in the manuscript. PLC refers to the broader category of liver carcinomas, whereas HCC is a specific type of PLC.

Thank you for the comment. We acknowledge the inconsistency in terminology and have standardized it by using “HCC” (Hepatocellular Carcinoma) throughout the manuscript.

Final remarks:

We thank the reviewer for evaluating our manuscript once again. We hope the revised manuscript can meet the standard of the esteemed journal.

Round 2

Reviewer 2 Report

Comments and Suggestions for Authors

1. Table 4: column 6, Please clarify I and C. It seems C stands for control instead of comparison.

Author Response

Dear Reviewer 2,

Thank you for the comment.

Response to Reviewer

Title:     Strategies for Pain Management in Hepatocellular Carcinoma Patients Undergoing Transarterial Chemoembolisation: A Scoping Review of Current Evidence

ID:          healthcare-3552969

Dear reviewer,

Thank you for reviewing our manuscript. We appreciate the constructive comments given and have responded to each of them as the following:

Comment

Reply

1. Table 4: column 6, Please clarify I and C. It seems C stands for control instead of comparison.

Thank you for the comment. We agree with you. We have revised it accordingly by replacing “Comparison” with “Control”:

Line 202: Table 4

Final remarks:

We thank the reviewer for evaluating our manuscript once again. We hope the revised manuscript can meet the standard of the esteemed journal.